# Sustainable Applications of Nanofibers in Agriculture and Water Treatment: A Review

Khandsuren Badgar [1],*, Neama Abdalla [2], Hassan El-Ramady [1,3] and József Prokisch [1]

[1] Faculty of Agricultural and Food Sciences and Environmental Management, Institute of Animal Science, Biotechnology and Nature Conservation, University of Debrecen, 138 Böszörményi Street, 4032 Debrecen, Hungary; hassan.elramady@agr.kfs.edu.eg (H.E.-R.); jprokisch@agr.unideb.hu (J.P.)

[2] Plant Biotechnology Department, Biotechnology Research Institute, National Research Centre, 33 El Buhouth Street, Dokki, Giza 12622, Egypt; neama_ncr@yahoo.com

[3] Soil and Water Department, Faculty of Agriculture, Kafrelsheikh University, Kafr El-Sheikh 33516, Egypt

* Correspondence: b_khandsuren@muls.edu.mn; Tel.: +36-203-413-997

**Abstract:** Natural fibers are an important source for producing polymers, which are highly applicable in their nanoform and could be used in very broad fields such as filtration for water/wastewater treatment, biomedicine, food packaging, harvesting, and storage of energy due to their high specific surface area. These natural nanofibers could be mainly produced through plants, animals, and minerals, as well as produced from agricultural wastes. For strengthening these natural fibers, they may reinforce with some substances such as nanomaterials. Natural or biofiber-reinforced bio-composites and nano–bio-composites are considered better than conventional composites. The sustainable application of nanofibers in agricultural sectors is a promising approach and may involve plant protection and its growth through encapsulating many bio-active molecules or agrochemicals (i.e., pesticides, phytohormones, and fertilizers) for smart delivery at the targeted sites. The food industry and processing also are very important applicable fields of nanofibers, particularly food packaging, which may include using nanofibers for active–intelligent food packaging, and food freshness indicators. The removal of pollutants from soil, water, and air is an urgent field for nanofibers due to their high efficiency. Many new approaches or applicable agro-fields for nanofibers are expected in the future, such as using nanofibers as the indicators for CO and $NH_3$. The role of nanofibers in the global fighting against COVID-19 may represent a crucial solution, particularly in producing face masks.

**Keywords:** natural fibers; cellulose; nano-medicine; agro-wastes; pollution; wastewater

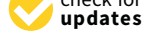



## 1. Introduction

Nanotechnology has become one of the most promising research fields at the beginning of the 21st century due to its leading technology of the new industrial revolutions. Due to the growing environmental concerns of natural fibers, they have occupied a great position in the research community, which have many advantages such as recyclability, cheaper, high specific properties, lower density or lightweight, and biodegradability [1,2]. Natural fibers also are considered renewable raw materials, which has become an obligatory issue for safe living [3]. The synthetic material-based composites should be replaced by natural ones in several manufacturing industries, such as the field of textiles, which flax was used in ancient Egypt nearly 7000 years ago [3]. The cellulose is the most important component of the lignocellulosic natural fibers, which many plants could order their content of cellulose (%) as follows straw of rice (41–57), leaf of date palm (46), leaf of abaca (56–63), bast of jute (61–71), leaf of banana (63–83), leaf of sisal (65), bast of hemp (68), bast of ramie (68.6–76.2), bast of flax (71), bast of kenaf (72), leaf of curaua (73.6), leaf of pineapple (81), bast of nettle (81–83), and seeds of cotton (83–91) [2,3].

Nanofibers, as nanoproducts, have been used in several sectors, including biomedical, pharmaceutical, agricultural, and industrial fields. Nanofibers could be found in natural and synthetic form, whereas nanocellulose and its derivative materials are common natural fibers. Nanofibers have many advantages, such as large specific surface area, high porosity, and high size uniformity, which allow applying nanofibers in many environmental issues such as wastewater treatment and removing pollutants from soils [1]. Nanofibers could be defined as nanostructures that may be fabricated using several methods such as the drawing method, template method, thermal-induced phase separation method, self-assembly, and electrospinning [4]. The electrospinning method is considered the best method (simple and easy) to fabricate non-woven nanofibers, which could be used due to the high-molecular-weight polymers [4]. Several applications of nanofibers have been confirmed, such as 3D printing of fiber-reinforced nanocomposites [5], suitable nanocomposite materials and fiber-reinforced polymer for airplane manufacture [6,7], fiber nano–bio-compositions for cranioplasty, and other orthopedic applications [8,9], replacing conventional rubber by the fiber-reinforced nanocomposites [10], using nano–coconut shell filler mixed jute mat-reinforced epoxy composites for reducing the weight of the structures [11].

Therefore, this review focuses on the difference between fibers and nanofibers, their applications, and their methods of fabrication especially using agro-wastes in producing nanofibers. The sustainable applications of nanofibers in the field of agriculture and the environment (particularly in wastewater treatment) are also discussed in the review.

## 2. Natural Fibers and Nanofibers

Agriculture is the main source of our supply of food, feed, fuel, and fiber. The natural fibers are produced by the harvested plants, where different fractions have these fibers (e.g., leaves, fruits, basts, stems, trunk, etc.) or animals (wool, silk, and hair) or minerals (asbestos). The sources of natural plant fibers may include plant leaves such as abaca, banana, and sisal; plant bast (e.g., kenaf, jute, and ramie, flax); seeds such as cotton; fruit such as coir and oil palm; plant straw (canola, maize, rice, wheat); and others [3,12]. More information about the natural fibers and differences between natural fibers and nanofibers can be found in Table 1. Nanofibers are defined as fibers whose diameters are in the nanometric range. Nanofibers have several main applications, including aerospace, 3D printing industry, orthopedic and structural applications, polyurethane matrix, paper, and textile industry [5,6,8–11,13–19]. It could also produce nanofibers from microbial sources such as bacterial cellulose, which could be used in antimicrobial, filtration, biosensor, gas sensor, and energy storage [20]. Bacterial cellulose nanofiber has been used for improving its recycled paper quality [21], removing hexavalent chromium [22], removing methylene blue [23], for 3D cell culture [24], or as a lithium-ion battery separator [25].

**Table 1.** A comparison between natural fibers and nanofiber, including their characterizations and main applications.

| Natural Fibers | Nanofibers |
|---|---|
| *Definition* | |
| The fiber is defined as a substrate of natural origin, which its length/diameter ratio is more than 1:200 [26] | *"Nanofibers could be defined as the fibers which have their diameters in nanometric range"* [27] |
| *The main sources* | |
| Green composites based on natural fibers compared to petroleum-based fiber composites [28,29] | Nanofibers are generally classified based on their composition into metal oxides, polymers, metals, carbon, ceramics, and hybrid [30] |
| *Main categories of natural fiber* | *Main types of nano-lignocellulose fibers* |
| 1—Mineral fibers (asbestos, basalt, and brucite) | 1—Lignocellulose nanofiber [31] |
| 2—Animal fibers (hair, silk, and wool) | 2—Bacterial nanocellulose [32] |
| 3—Plant fibers (lignocellulose) [33] | 3—Nanocrystalline cellulose [34] |
| | 4—Nano-fibrillated cellulose [16] |
| *Main treatments for natural fibers* | *Main fabrication techniques of nanofibers* |
| Chemical (acetylation, alkaline, benzoylation, peroxide, potassium permanganate, silane, and stearic acid) and surface treatments [1] | Non-electrospinning techniques (i.e., phase separation, drawing, template synthesis, and self-assembly), electrospinning, and hydrothermal techniques [35–38] |
| *The main applications of natural fibers* | *The main applications of nanofibers* |
| Automobile, construction, aerospace, and marine structural industries [39,40] | Aerospace, 3D printing industry, orthopedic and structural applications, polyurethane matrix, paper and textile industry [13–15] |

## 3. Producing Nanofibers from Agro-Wastes

A large number of discarded wastes of many crops every year was produced along with the crop production, which may cause environmental problems and have potential safety hazards [41]. Globally, about 5 billion tons of bio-wastes are generated annually from agricultural activities [42]. These wastes are the main source of a huge amount of cellulose every year, such as banana rachis, coconut husk fiber, grain straw, grape skin, garlic peel, soy hulls, and sugarcane bagasse [43]. These amounts of wastes could also be produced from both the agriculture and forestry industries, which are characterized as renewable, biodegradable, and low raw material cost. Cellulose nanofibers could be divided into cellulose nanofibrils and cellulose nanocrystals (diameter 5–30 and 3–10 nm, respectively), which are described as flexible, long, rope-like fibers with both crystalline and amorphous regions [44]. Several studies were published concerning the use of different agro-wastes in producing the nanofibers such as wastes of coconut husk and rice husk [43], wastes of pineapple leaves [45], sugarcane bagasse [46], Eucalyptus sawdust [47], wastes obtained from orange juice processing [48], quinoa wastes [49], discarded wooden bark of Kozo plant [50], pomegranate peel [51], and wastes of peach branches [41]. A survey of the most recent published articles regarding the use of agro-wastes in producing nanofibers is listed in Table 2.

**Table 2.** List of some different agro-wastes that are used in producing nanofibers.

| Nanofibers Obtained from Agro-Wastes and Used Method | Comment on Nanofibers | References |
|---|---|---|
| Polyvinyl alcohol/starch nanocomposite film reinforced with cellulose nanofiber of sugarcane bagasse was produced using alkaline acid treatment under ultrasonication | Nanocomposite film reinforced with cellulose nanofiber | [46] |
| Using wastes of bamboo (*Phyllostachys pubescens*) as lignocellulosic biomass using microwave-assisted ethanol solvent treatment to produce cellulose nanofiber | Cellulose nanofibers | [52] |
| Lignocellulosic nanofiber can be produced by washing the *Eucalyptus* sawdust with an aqueous surfactant solution | Bio-nanocomposite films | [47] |
| Wastes obtained from orange juice processing can be used to obtain biodegradable film of reinforced cellulose nanofiber | Nano-biocomposite films | [48] |
| Using pomegranate (*Punica granatum* L.) peel extract beside polyvinylpyrrolidone and polyvinyl alcohol | Nanofibers for cosmeceutical purposes | [53] |
| Quinoa wastes incorporated with multi-walled C-nano tubes-ZnO can be used to obtain natural cellulose fibers | Bio-nanocomposite | [49] |
| Producing cellulose nanofibers obtained from the discarded wooden bark of Kozo plant by acidified sodium chlorite and acetic acid | Cellulose nanofibers | [50] |
| In vitro assay of nanofibers obtained from ethanolic extract of pomegranate peel used electrospinning method | Gelatin nanofiber | [51] |
| Peach branches used under high-pressure homogeneous to produce peach branches–cellulose nanofiber | Nanofiber reinforcer of gelatin hydrogel | [41] |
| Crystalline nanocellulose was generated using coconut husk, and rice husk by hydrolysis disintegration | Mechanically reinforced polymer composites | [43] |
| Nanocellulose incorporated in poly-lactic acid matrix obtained from cotton wastes by acid hydrolysis | Production of nanocellulose | [54] |
| Producing cellulose nanofiber from pineapple leaf wastes, which reinforced into a polystyrene substrate | Cellulose nanofiber reinforced polystyrene nanocomposites | [45] |

## 4. Applications of Nanofibers in Agriculture

Recently, many researchers studied the main applications of nanofibers in agriculture because of their tailoring properties, including the biocompatible and biodegradable features, high surface area and porosity, ease of active ingredient additions (i.e., fungicides, insecticides, herbicides, pesticides, hormones, and pheromones), and flexibility of electrospun nanofibers [55]. Nanofibers can apply for plant protection (through applying pesticides for pest control), plant growth (through applying hormones and/or fertilizers), pollution and contamination controls, and irrigation systems (through water filtration), as reported in Table 3 by Meraz-Dávila et al. [56], Raja et al. [57], and [58].

**Table 3.** The main applications of nanofibers in the agricultural sectors as reported by the literature.

| Main Applications of Nanofibers in Agricultural Sectors | References |
|---|---|
| 1—Nanofibers for good germination by coating seeds | [59–62] |
| 2—Agro-wastes for production nanofibers | [63,64] |
| 3—Nanofibers-based filters for irrigation systems | [65] |
| 4—Nanofibers for plant protection | [56] |
| 4.1 Encapsulation of fungicides | [66,67] |
| 4.2 Encapsulation of herbicides | [68] |
| 4.3 Detecting trace pesticides in water | [69] |
| 5—Nano-silica grafted fiber | [70] |
| 6—Smart nanotextiles for sustainable agriculture | [13] |
| 7—Nanofibers for encapsulation of agrochemicals | [71,72] |
| 7.1 Fertilizer application | [73] |
| 7.2 Plant hormones (e.g., indole acetic acid) | [57,74] |

The main applications of nanofibers in the agricultural field may include coating seeds [60–62], nanofibers-based filters for irrigation systems [65], nanofibers for plant protection [56] through encapsulation of fungicides [66,67], or detecting trace some pesticides in water [69], nano-silica grafted fiber [70], smart nanotextiles for sustainable agriculture [13], nanofibers for encapsulation of agrochemicals including fertilizer [75], and phytohormones [71,72]. Nanofibers can be used as a smart and sustained delivery of agricultural inputs through seed to improve germination and seedling growth in rice [59,76] and cowpea [60], groundnut [57], and sesame [62].

Nanofibers were applied for plant protection through encapsulation of pesticides [56], including fungicides [66,67], herbicides [68], nano-silica grafted fiber [70], and smart nanotextiles for sustainable agriculture [13]. The use of nanofibrous filters in irrigation systems may involve functionalization (i.e., adsorption, filtration, and sterilization) by bioactive compounds, which could be achieved by interfacial polymerization, doping nanoparticles, self-assembly, and surface coating cross-linking or grafting, layer-by-layer [27]. Several nanomaterials such as graphene oxide could be used for water purification because of its multi-functionality, such as an antibacterial agent, excellent adsorption property, and photocatalytic abilities [77]. Thus, nanofibers could be sustainably applied in many agricultural processes that lead to reduce the loss in used agrochemicals pesticides, hormones, and/or fertilizers [57,61–74], and to increase the productivity of crops through innovative management of phytopathogens or nutrients [13].

## 5. Nanofibers for Water/Wastewater Treatment

Nanofibers are considered promising tools that are applied for diverse environmental conditions, especially polysaccharide-based electrospun nanofibers. The groups of polysaccharides are suitable materials for these environmental issues because of their biobased origins, variety of types, eco-friendly, and renewable nature [78]. In general, the nanofibers have been applied for many environmental problems such as removing pollutants from the air by filtration [79], water treatment [80], antimicrobial treatment [81], environmental sensing [82], for heavy metal removing as adsorbents [83,84], and agricultural/environmental remediation [78,85]. The environmental sustainability of water using cellulose nanofibers-based green nanocomposites is considered one of the most important environmental issues [86].

Based on the potential of water treatments under the global water crisis in a pure and safe case, using nanofibers in this review in water/wastewater treatment and removing the pollutants is discussed in more detail as an urgent environmental task (Table 4). The pollution of water causes an imbalance in different ecological environments and directly also affects human health. Thus, there is a great need for researchers to develop effective technology in wastewater purification [35]. The main strategies in wastewater

treatment may include filtration, adsorption, catalysis, centrifugation, biological treatment, and electro-coalescence [87]. More than 200 natural and synthetic polymers were successfully electrospun into nanofiber membranes, such as polyimide (PI), polyacrylonitrile (PAN), poly/vinyl alcohol (PVA), poly/vinylidene-fluoride (PVDF), polylactic acid (PLA), cellulose acetate (CA), polyurethane (PU), polyethylene oxide (PEO), and polycaprolactone (PCL) [35].

**Table 4.** Using of nanofibers in water/wastewater treatments for removing heavy metal pollutants.

| Nanofibers and Their Average Diameter | Max. Adsorption Capacity | Pollutant | References |
|---|---|---|---|
| Polyvinylidene fluoride–polyacrylonitrile-ZnO nanofiber membranes (200 nm) | 350 mg g$^{-1}$ | Cd | [88] |
| Amidoxylated polyacrylonitrile/Poly-vinylidene fluoride (AOPAN/PVDF) (235–314 nm) | 89.29 mg g$^{-1}$ | Pb (II) | [89] |
| Nitro-oxidized carboxy-cellulose nanofibers obtained from moringa plants (0.22 μm) | 257.07 mg g$^{-1}$ | Hg | [90] |
| Electrospun chitosan–polyethylene oxide-oxidized cellulose biobased composite (159.3 nm and 21.7 μm, resp.) | 15.72 mg g$^{-1}$ | Cu | [91] |
| Modified poly butylene succinate nanofibers (10 μm) | 91.2 and 122 mg g$^{-1}$, respectively | Ag (I) and Hg (II) | [92] |
| TEMPO-oxidized cellulose nanofibers (diameter 6.15 nm) | 56.50 mg g$^{-1}$ | Cu (II) | [93] |
| Polyvinyl alcohol (PVP)-octa-amino-POSS nanofibers (21 μm) | 37.4 and 120 mg g$^{-1}$, respectively | Cu (II), Pb (II) | [94] |
| Starch-g-poly(acrylic acid)-cellulose nanofiber bio-nanocomposite hydrogel (10 μm) | 40.65 mg g$^{-1}$ | Cd (II) | [95] |
| Oxidized regenerated cellulose nanofiber membrane (10 μm) | 20.78 and 206.1 mg g$^{-1}$, respectively | Cu (II), Pb (II) | [96] |
| polyvinylidene fluoride–amidoximized polyacrylonitrile nanofibers (20.7 μm) | 30.1, 25.8, and 72.5 mg g$^{-1}$, respectively | Cu (II), Ni (II), Pb (II) | [97] |
| Modified prepared polyacrylonitrile nanofibers (320 nm) | 22.95 and 12.36 mmol g$^{-1}$, respectively | Cu and Pb | [98] |
| Centrifugal spinning of lignin amine/cellulose acetate nanofiber (756 nm) | 50.08 and 31.17 mg g$^{-1}$, respectively | Cu (II), Co (II) | [83] |
| Visualized chitosan–polyacrylonitrile nanofiber membrane | 164.3 mg g$^{-1}$ | Cu (II) | [99] |
| Zn/Al/gallate layered double hydroxide–polystyrene nanofibers (2–5 μm) | 190 mg g$^{-1}$ | Cu (II) | [100] |
| Polyacrylonitrile–polyetherimide nanofibers (0.84 mm) | 242.7, 214.1, 258.3 mg g$^{-1}$, respectively | Cu (II), Cr (VI), As (V) | [101] |

Polyhedral Oligomeric Silsesquioxane (POSS); 2,2,6,6-tetramethylpiperidine-1-oxyl (TEMPO); Amidoxylated polyacrylonitrile (AOPAN).

Concerning the mechanism of cellulose nanofibers (CNFs) in water purification, CNFs can make a link with carboxylic surface functional groups by oxidation and chemically bonded nanocomposites based on modified CNFs with various metal–organic or metal pillars frameworks in order to create a robust and high-efficiency material [86]. The mechanism of water purification using nanofibers consists of both physical and chemical methods that could be explained based on the chemical and physical bases (Figure 1). Chemically, the formation of stable chemical bonds between nanofibers and metal ions and relevant oxidation-reduction is involved. Physically, the surface area and pore volume of nanofibers are key parameters determining the fiber adsorption capacity and, therefore, water treatment performance. This mechanism was confirmed by many researchers, such as Agrawal et al. [102] and Uddin et al. [103]. Many recent reviews were published on the removing of hazardous pollutants (i.e., both organic and inorganic materials) from water/wastewater using nanofibers such as Chen et al. [104], Cui et al. [35], Ibrahim et al. [105], Jahan and Zhang [106], Marinho et al. [107], Sakib et al. [80], Sjahro et al. [108], and El-Aswar et al. [109]. These previous studies confirmed that electrospun nanofiber membranes could be easily used for achieving different water treatments by combining multifunctional materials due to their high specific surface area and unique interconnected structure [104].

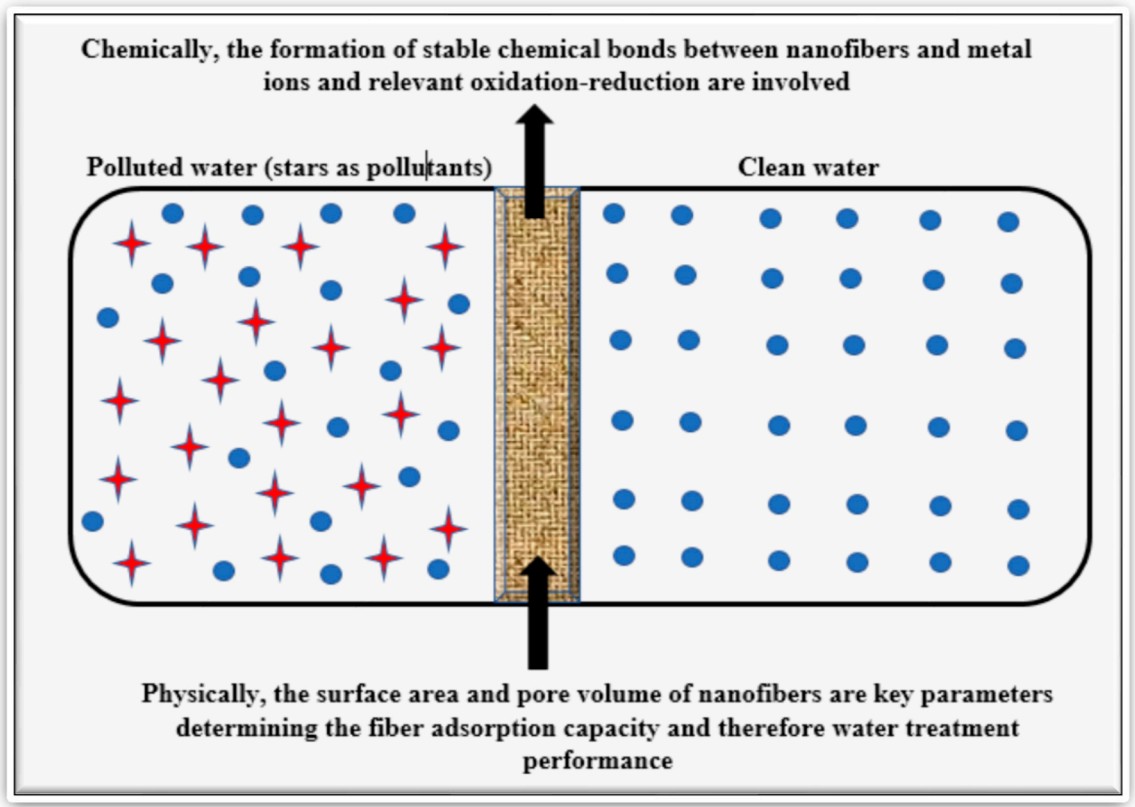

**Figure 1.** The mechanism of water purification using nanofibers consists of both physical and chemical methods, which illustrates electrostatic and intermolecular forces between nanofibers and pollutants in water.

### 6. Nanofibers for Food Packaging

In the food industry, nanofibers were applied in many processes such as encapsulation of food materials [110], food preservation [111], beverage industry [112], and food packaging industry [113]. Food packaging is considered a powerful impact on maintaining food safety and its quality [113] through some functional properties, including gas or moisture absorbents, specific gas barrier, antimicrobial properties, antioxidant activity, UV protection, or monitoring capacity to report product quality [114]. The packaging materials are important materials that could use in minimizing bacterial and chemical spoilage of foods [115]. Currently, food packaging can be prepared into intelligent and active forms [116]. Active packaging is maintaining or improving the packaged food conditions or extending its shelf-life, whereas intelligent packaging is the monitoring of the packaged food conditions or the surrounded environment of the foods [114]. The group of intelligent packaging systems is composed of colorimetric indicators, which may provide essential information on any changes occurring in a food product or its surroundings, such as temperature or pH, through observing visual color changes [113] (see Figure 1).

Due to their low price, availability, and desirable properties, petroleum-based plastics are widely used as food packaging materials. These such plastic materials are non-biodegradable, causing many environmental problems reducing food safety due to the migration of some compounds such as monomers, plasticizers, and solvent residues from plastics into the food [48]. Functionalized nanomaterials driven antimicrobial food packaging are considered promising solutions as alternative substances for food packing [117]. The sustainable and green sources for producing edible films for food packaging have been widely used, such as many fruit and vegetable purees as wastes resulting from food processing [48]. Thus, nano-based filler compounds, including cellulose nanofibers, nano-clay,

and nanometals, were utilized to improve the mechanical, physical, and gas inhibitory properties of edible films (Table 5).

**Table 5.** The main applications of nanofibers in the field of food industry and packaging.

| Main Applications According to Different Food Processes and Industry | References |
| --- | --- |
| 1—Nanofibers for the field of food industry | [118] |
| 2—Nanofibers for beverage industry | [112] |
| 3—Nanofibers for encapsulation of food materials | [110] |
| 4—Nanofibers for food preservation | [111] |
| 5—Nanofibers for food packaging industry | |
|     5.1 Nanofibers for food intelligent packaging | [113] |
|     5.2 Nanofibers as an active food packaging system | [119] |
|     5.3 Nanofibers for active–intelligent food packaging | [120] |
|     5.4 Nanofibers containing biodegradable polymers | [121] |
|     5.5 Nanofiber for active food packaging | [122] |
|     5.6 Nanofibers for food freshness indicators | [113] |
|     5.7 Cellulose-based hydrophobic materials for food packaging | [123] |
|     5.8 Functionalized nanomaterials driven antimicrobial food packaging | [117] |

Nanofibers such as cellulose nanofibers have been reinforcement agents in recent years that have high thermal and chemical stability compared with other organic nanoparticles. Due to the impermeability of nanofillers, cellulose nanofiber can trigger a controlled release of active compounds and form complex diffusion pathways [48]. Accordingly, packaging films can control pathogens and improve the quality and shelf life of food by acting as carriers of antimicrobial, antioxidant, and active compounds [48]. The use of nanofibers in food packaging may include different topics as presented in Table 5, which focused on using nanofibers for active–intelligent food packaging [113,120,122], as an active food packaging system [97], nanofibers containing biodegradable polymers [121], nanofibers for food freshness indicators [113], cellulose-based hydrophobic materials for food packaging [123], nanofibers for electrochemical DNA biosensors [124], and functionalized nanomaterials driven antimicrobial food packaging [117].

## 7. Nanofibers for Biomedical Fields

The most common method in producing nanofiber is polyvinyl alcohol (PVA), as a good candidate to be used for plastic and packaging materials to produce synthetic biodegradable polymers (Figures 2 and 3). A polymer solution of electrospinning depends on many factors such as polymer characterization (solubility and its molecular weight), solvent type (volatility, vapor pressure, and dielectric constant), polymer solution (viscosity and surface tension), and ambient conditions, including temperature and relative humidity [107]. Several advantages of PVA were reported as excellent film-forming, adhesive, and emulsifying properties, which could be used in different industries through its potential as an agent for sizing of textile, an adhesive of paper, and soluble packaging films [46]. However, some limits in PVP usage were also reported, including its high cost, poor moisture barrier, and low biodegradation rate, which need to modify PVA using other polymers (such as starch and gelatin) and nanoparticles [121]. Due to the low mechanical strength of the biodegradable polymers alone obtained from PVP, some materials are needed to be inserted to improve these properties of biodegradable plastic [46]. Many applications of the synthesis of natural cellulose nanofibers could be explained in the following sub-sections.

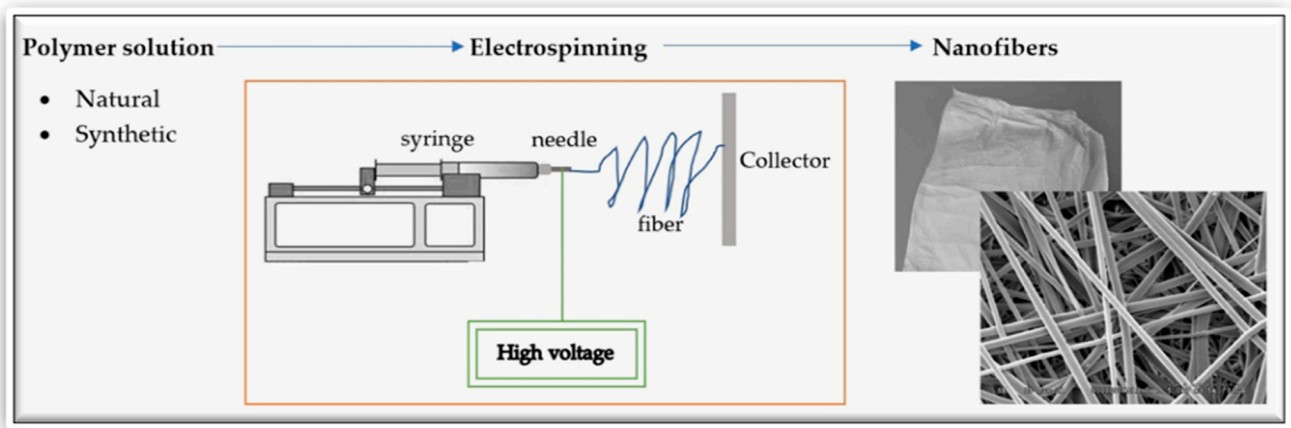

**Figure 2.** A schematic process of horizontal electrospinning setup. Nanofibrous membrane was produced from 10% of polyvinyl butyral polymer by horizontal electrospinning setup.

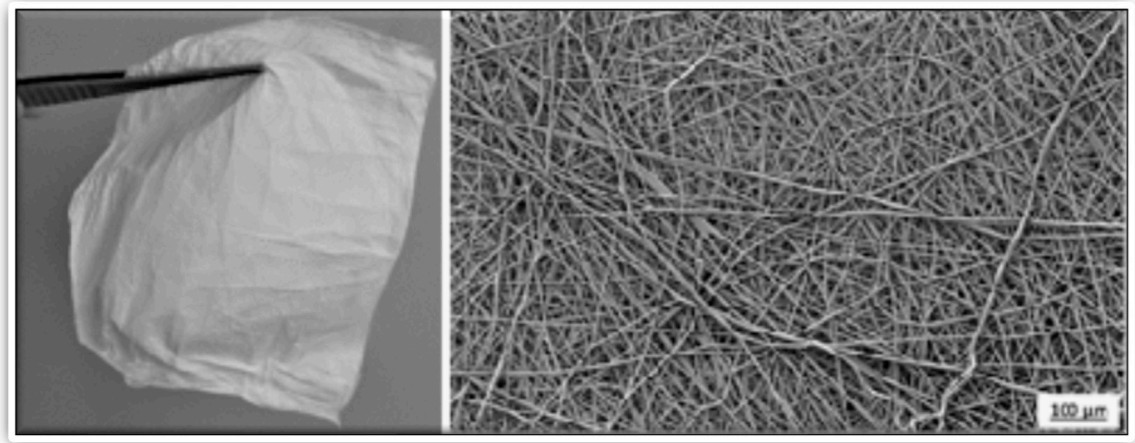

**Figure 3.** Optical and SEM image of nanofibrous membrane.

The application of nanofibers or with some bioactive compounds is increasing in medicine as drug release capsule; transdermal absorbent; drug release artificial skin; wound dressing; artificial blood vessel; covered stent; artificial cornea; artificial skin; filling agent for artificial bone, wound, and therapeutic applications; nerve or organ patch wound covering and protective agent; dialysis membrane; surgical adhesive sheet; and adhesion prevention materials [63]. Especially, nanofibers are still the most widely used as drug delivery control systems due to their functional properties, ability to deliver drugs to specific targets, increased surface area, improved dissolution rate of specific drugs [125], and ability to encapsulate various drugs. The medical applications of nanofibers were investigated primarily in humans, but some researchers are considering them for veterinary applications. However, there is a limited result on the practical application of electrospun nanofibers in veterinary medicine. Poly(3-hydroxybutyrate-co-3-hydroxyvalerate) (PHBV) electrospun fibers were filled with up to 20% of diclofenac sodium for encapsulation and release, which is recommended for humans and animals for pain, fever, and inflammation [126]. Another study presented that zein electrospun nanofibers were loaded with progesterone are ideal media for progesterone delivery for bovine estrus synchronization. In particular, nanofibers loaded with 1.2 g of progesterone had much stronger control of progesterone delivery than 1.9 g and 2.5 g of progesterone, releasing 87.28% of progesterone by 7 days [127].

Several studies reported about different applications of the nanofibers in medical sectors, such as Malik et al. [37], de Carvalho and Conte-Junior [128], Ghajarieha et al. [129], Karthega et al. [130], Rivelli et al. [131], Urbina et al. [63], Zhao et al. [132] and Zhong et al. [133], as well

as in the pharmaceutical applications, for example, Imani et al. [134], Balusamy et al. [135], Dodero et al. [136], Kumar et al. [137], Haidar et al. [138], and Pandey [139], as listed in Table 6. The healing process of bones and tendons is very slow, and depending on the severity of the injures, it takes at least six weeks or more for them to regain normal alignment. The slow healing process of injured tissue in the tendon is mainly caused by an inadequate blood supply. Deep tendon tissue receives the nutrients it needs, probably through diffusion. Some studies reported that nanofiber-based scaffolds with functional additives have a greater ability to treat these problems in animals.

**Table 6.** A list of applications of nanofibers in biomedical ad pharmaceutical fields.

| Applications of Nanofibers in Medicine | Applications of Nanofibers in <break/>Pharmacology |
|---|---|
| Adhesion prevention materials | Anticancer drug delivery |
| Artificial blood vessels, cornea, and skin | Antimicrobial drug delivery |
| Drug release capsule | Antibiotic drug delivery |
| Drug release artificial skin | Anti-inflammatory drugs |
| Dialysis membrane | Cell delivery and tissue engineering |
| Facemask, skin and vascular tissue engineering | Growth factor and protein delivery |
| Nerve or organ patch | Neuroprotective drugs |
| Rhinosinusitis treatment | Nucleic acid delivery |
| Surgical adhesive sheet | Miscellaneous drug delivery |
| Transdermal absorbent | Controlled release of gentamicin |
| Wound covering and protective agent | Localized chemotherapy |
| Filling agent for artificial bone | Smart active drug release systems |
| Wound dressing and healing systems | Transdermal drug delivery |
| Wound and therapeutic applications | Double-layered planar nanofibrous scaffolds abdominal adhesion prevention |

Poly-L-lactic acid (PLLA) nanofiber membrane was used in the treatment of bone damage in rabbit tibia and compared with porous collagen membranes and collagenous membranes reinforced by nanofiber membranes. After 3 weeks of the treatment, bone tissue formation was high in the collagenous membrane reinforced by the nanofiber membranes treated group. After 6 weeks of the treatment, the regeneration of cortical bone tissue was also better in the collagenous membrane reinforced by nanofiber membranes treated group than other groups. Additionally, nanofiber membrane and porous collagen membrane treated groups were filled with spongy bone-like tissue during the treatment. These results were indicated that electrospun nanofibers or combined with collagen could improve bone regeneration [140]. Similarly, other studies reported that the effect of other polymer nanofibers with active compounds in bone damage.

Poly(lactide-co-glycolide) scaffolds were enriched with calcium phosphate nanoparticles (PLGA/CaP) and silver doped calcium phosphate nanoparticles (PLGA/Ag-CaP), and they were shown great biocompatibility and bone healing without being absorbed by adjacent bone during the treatment of bone defects in sheep. Both scaffolds are allowed bone formation directly at the center of the former defect, and surface integrals of new bone formation were very similar [141]. Another study reported that electrospun cellulose–iron acetate nanofibers were also enhanced osteoblast cell attachment and proliferation among mats' porous [142].

Polyvinyl alcohol (PVA) nanofiber contained Eucalyptus globules extract was used in the treatment of Achilles tendon injures. As a result, nanofibers loaded with eucalyptus globules extract were shown reduced new angiogenesis, increase the ratio of fibroblasts to fibroblasts, reduce edema, and the low placement of fine collagen fibers during treatment compared to the untreated group [143]. The treatment mechanism is explained by inhibiting oxidative stress. Eucalyptus extract is contained flavonoid and phenolic compounds,

including quercetin, tannins, and saponins, which have antioxidant, antimicrobial, and anti-inflammatory activities [144]. Basically, if nanofibers contain antioxidant agents, they will have a potent ability to reduce the progression of oxidative stress by controlling reactive oxygen species (ROS).

The mechanical properties and biocompatibility of electrospun nanofibrous membrane are very promising in tissue engineering for implantation in the injures including vascular, skin, cartilage, etc. The idea of combining different structures and materials with building a multi-component tissue engineering vascular scaffold to improve mechanical properties and biocompatibility was intensively utilized [145,146]. Electrospinning has emerged as a common technique for producing nanofiber-based scaffolds for vascular and endothelial reconstruction. This is because the simulation of the microstructure of natural arteries and integration of the graft with surrounding cells and tissues are available [147,148]. Furthermore, it is convenient to incorporate the regulatory components into the drafts during the electrospinning process [149,150].

The hybrid small-diameter vascular graft with sustained heparin release was developed from electrospun poly($\varepsilon$-caprolactone) (PCL) and chitosan, which was shown the anti-thrombogenic and endothelialization properties. Heparin functionalization in nanofibers clearly improved the blood compatibility of these vascular grafts. It was continuously released from the graft for over a month [150]. In addition, in vitro study investigated the cartilage formation of mesenchymal stem cells (MSCs) on PCL nanofiber scaffolds in the presence of transforming growth factor-beta (TGF-$\beta$1). The differentiation of stem cells into chondrocytes on the nanofiber framework was comparable to established cell pellet cultures. It was advantageous to use nanofibers instead of the cell pellet system for better mechanical properties, oxygen/nutrient exchange, and ease of manufacture. The authors reported that the PCL nanofiber scaffold is practical support for MSC transplantation and is a candidate scaffold for a cell-based tissue engineering approach to cartilage repair [151].

A femoral artery model in canine was developed from poly(L-lactide-co-caprolactone) (PLCL) scaffolds with collagen/chitosan in a 3:1 ratio. The draft scaffolds were shown improved long-term patency, better growth of endothelial cells (ECs) and smooth muscle cells (SMCs), and improved vascular gene and protein expression compared to non-concentrated scaffolds [152]. In vitro study, PLCL nanofiber-based scaffolds with tussah silk fibroin as a vascular draft effectively promoted the adhesion and proliferation of vascular endothelial cells [153]. Moreover, electrospun nanofiber scaffolds for vascular graft application were proposed by an in vitro study that allowed the mechanical properties of vascular substitutes to be adjusted, and compliance adjustment for vascular tissue engineering improved [154].

A functional 3D model of stromal and epithelial cells was developed from polyglycolic acid (PGA) nanofiber-based scaffolds and used to construct a functional reconstitution of bovine endometrium. In a typical procedure, stromal cells were seeded into the scaffold first, followed by epithelial cells after 1 or 7 days. The epithelial cells seeded on day 1 were represented more natural endometrial tissue than the epithelial cells seeded on day 7. The epithelial and stromal cells co-cultured on the scaffold were showed proper expression of the natural endometrium and ZO-1 cytokeratin and vimentin by the epithelial cells [155]. Basically, in vitro and in vivo studies confirmed that PGA nanofiber-based scaffolds or with some additives and their essential properties in medical applications. Xu and co-workers that PGA scaffolds containing cells derived from tendon sheaths are functionally and structurally similar to natural sheaths. After 12 weeks of surgery, this artificial sheath formed a relatively mature structure and had a smooth inner surface and the histological structure of the well-developed sheath with a clear space between the tendon and the artificial sheath. Surprisingly, compared to the tendons that were surrounded by scar-repaired tissue, the tendons used less energy to slide in the constructed sheath [156]. In addition, PGA scaffolds containing 10% and 30% of gelatin improved the endothelial cells and smooth muscle cell adhesion and survival [157].

After spreading the coronavirus disease (COVID-19), many approaches have been applied to prevent this pandemic, including the use of respirators, practicing personal

hygiene, social distancing, and wearing face masks [158]. Electrospun nanofibers are considered suitable air filtration devices (diameters range from 40 to 2000 nm), which allow air and prevent microbes from passing [159]. Thus, many recent publications have been focused on the potential of "electrospun nanofibers-based face masks" to reduce the spreading of SARS-CoV-2 among humans because these nanofibers may achieve virus blocking, antivirus selectivity, biodegradability, etc. [160–162].

## 8. Conclusions

A comparison between natural fibers and nanofibers was discussed in this review, besides different sustainable applications of nanofibers in fields of biomedicine, agriculture, and the environment. Producing nanofibers from different agro-wastes and applying nanofibers for food packaging were also the main topics in this manuscript. Based on the environmental problems of synthetic nanofibers, particularly petroleum-based fiber composites, natural nanofibers are recommended especially after reinforcement, along with the matrix, for more performance and strength of the composites. The nano–bio-composites could produce by coupling matrix of nanoparticles into bio-reinforcer, which converted into the biofiber-reinforced polymer matrix. Nano-based reinforced polymeric composites can be applied for sophisticated applications, mainly in agriculture (irrigation system, seed coating, for plant protection, agrochemical encapsulation), environmental (removing pollutants from the air by filtration, antimicrobial treatment, environmental sensing, for heavy metal removing as adsorbents, and agricultural/environmental remediation, and wastewater treatment), and food sectors (food packaging, beverage industry, encapsulation of food materials, food preservation, and nanofibers for electrochemical DNA biosensors). From the previous applications, using nanofibers in water treatment and food packaging are important areas for research and development. Several further investigations are needed to answer questions concerning the sustainable application of nanofibers in our life, i.e., what are the new approaches for using nanofibers in agriculture? Can we use the nanofibers as indicators for CO and $NH_3$ in the air of rooms? Can we use nanofibers in nano-biofortification? What are the expected roles of nanofibers-based face masks in the global fighting against COVID-19?

**Author Contributions:** J.P. and H.E.-R. developed the idea and outline of the review. K.B. wrote the biomedical section. N.A. wrote the agricultural section of the manuscript, whereas the rest of the sections were written by H.E.-R. and K.B., and N.A. revised the manuscript thoroughly and finalized it. All authors have read and agreed to the published version of the manuscript.

**Funding:** H. El-Ramady thanks the Central Department of Mission, Egyptian Ministry of Higher Education (Mission 19/2020), and also the Hungarian Tempus Public Foundation (TPF), grant no. AK-00152-002/2021 for financing and supporting this work.

**Institutional Review Board Statement:** Not applicable.

**Informed Consent Statement:** Not applicable.

**Conflicts of Interest:** The authors declare no conflict of interest.

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
