# Peer review of "Sustainable Applications of Nanofibers in Agriculture and Water Treatment: A Review"

_sustainability, doi:10.3390/su14010464_

Round 1

Reviewer 1 Report

This review written by Khandsuren Badgar et al is a timely summary on the agriculture, wastewater treatment and biomedicine application of a variety of nanofibers and their composites that have been reported mostly over the last two years. My suggestion therefore is to publish the work after the authors addressing the following concerns:

1. The mechanism of water purification using nanofibers consists of both physical and chemical methods. Here a schematic figure is necessary to illustrate electrostatic and intermolecular forces between nanofibers and pollutent in water.  Chemically, the formation of stable chemical bonds between nanofibers and metal ions and relevant oxidation-reduction are involved. Physically, the surface area and pore volume of nanofibers are key parameters determining the fiber adsorption capacity and therefore water treatment performance.

2. "7 Nanofibers for biomedical fields” 
In this section, more possibilities with respect to biomedical application using electrospun nanofibers
can be included. Face masks, protective clothing  and ultrafine fiber filters with suitable composition and adjustable nanostructures will achieve virus blocking, antivirus selectivity,  biodegradability, etc.

3. There are only two figures in the manuscirpt, yet both are labeled as Figure 1. 

4. The statements in the first paragraph of 'Conclusions' might be better moved to 'Introduction'. Meanwhile, conclusive remarks and future directions regarding sustainable applications of nanofibers are encouraged to be included here. 

Author Response

Reviewer 1#

This review written by Khandsuren Badgar et al is a timely summary on the agriculture, wastewater treatment and biomedicine application of a variety of nanofibers and their composites that have been reported mostly over the last two years. My suggestion therefore is to publish the work after the authors addressing the following concerns:

The Response: thanks!

  1. The mechanism of water purification using nanofibers consists of both physical and chemical methods. Here a schematic figure is necessary to illustrate electrostatic and intermolecular forces between nanofibers and pollutant in water.  Chemically, the formation of stable chemical bonds between nanofibers and metal ions and relevant oxidation-reduction are involved. Physically, the surface area and pore volume of nanofibers are key parameters determining the fiber adsorption capacity and therefore water treatment performance.

The Response: Done, thanks!

  1. "7 Nanofibers for biomedical fields” 
    In this section, more possibilities with respect to biomedical application using electrospun nanofibers can be included. Face masks, protective clothing and ultrafine fiber filters with suitable composition and adjustable nanostructures will achieve virus blocking, antivirus selectivity, biodegradability, etc.

The Response: a part including these themes was added, thanks! (Ln: 313-318)

  1. There are only two figures in the manuscript, yet both are labeled as Figure 1.

The Response: Corrected, thanks!

  1. The statements in the first paragraph of 'Conclusions' might be better moved to 'Introduction'. Meanwhile, conclusive remarks and future directions regarding sustainable applications of nanofibers are encouraged to be included here.

The Response: changed in both introduction and conclusion sections, thanks! (Ln: 51-55)

Reviewer 2 Report

It is very difficult to understand each descriptions in all tables. Surroundings with ruled lines are essential to read.

Is there any limitation for width of table?
Author used more width in Figure 1 (There are two figure 1 on page 8).

Author Response

Reviewer 2#

It is very difficult to understand each description in all tables. Surroundings with ruled lines are essential to read. Is there any limitation for width of table?

The Response: adjusted and corrected, thanks

Author used more width in Figure 1 (There are two figure 1 on page 8).

The Response: corrected, thanks!

Reviewer 3 Report

The review covers the applications of natural fibres in different segments.  Although the review has no fixed boundaries as a reviewer, there are some concerns that need to address before considering the work for publication:

  1. The correction of the English language for a better flow of reading is a must.
  2. Please try to include microbial cellulose fibres in your study.
  3. Please try to mention the novelty of this review more clearly.
  4. Table 1: Please check the definition of Natural Fibres (should it contain synthetic origin?). The idea from the main sources doesn't fit natural fibres.
  5. Table 2, point 2: please make it more clear.
  6. Line 157: If the membranes have high porosity how can they perform better purification? Please elaborate.
  7. Line 177-178: Please complete the sentence.
  8. Section 6: Please restructure. Keep focus and elaborate on the basic idea.

Author Response

Reviewer 3#

The review covers the applications of natural fibres in different segments.  Although the review has no fixed boundaries as a reviewer, there are some concerns that need to address before considering the work for publication:

  1. The correction of the English language for a better flow of reading is a must.

The Response: OK, thanks!

2. Please try to include microbial cellulose fibres in your study.

The Response: added, thanks! (Ln: 75-79)

3. Please try to mention the novelty of this review more clearly.

The Response: ok, done, thanks (Ln: 29-32)

4. Table 1: Please check the definition of Natural Fibres (should it contain synthetic origin?). The idea from the main sources doesn't fit natural fibres.

The Response: ok, done, thanks

5. Table 2, point 2: please make it more clear.

The Response: ok, done, thanks

6. Line 157: If the membranes have high porosity how can they perform better purification? Please elaborate.

The Response: ok, corrected, thanks

7. Line 177-178: Please complete the sentence.

The Response: ok, completed, thanks!

8. Section 6: Please restructure. Keep focus and elaborate on the basic idea.

The Response: Section 6 in Table 5 removed from the Table and inserted into the text, thanks!